# An Overview of the Factors Involved in Biofilm Production by the *Enterococcus* Genus

**DOI:** 10.3390/ijms241411577

**Published:** 2023-07-18

**Authors:** Pavel Șchiopu, Dan Alexandru Toc, Ioana Alina Colosi, Carmen Costache, Giuseppe Ruospo, George Berar, Ștefan-Gabriel Gălbău, Alexandra Cristina Ghilea, Alexandru Botan, Adrian-Gabriel Pană, Vlad Sever Neculicioiu, Doina Adina Todea

**Affiliations:** 1Department of Microbiology, “Iuliu Hațieganu” University of Medicine and Pharmacy, 400012 Cluj-Napoca, Romania; 2Department of Pneumology, “Iuliu Hațieganu” University of Medicine and Pharmacy, 400332 Cluj-Napoca, Romania; 3Faculty of Medicine, “Iuliu Hațieganu” University of Medicine and Pharmacy, 400012 Cluj-Napoca, Romania

**Keywords:** enterococcus, biofilm formation, antibiotic resistance, quorum sensing, virulence factors, extracellular DNA (eDNA), pheromone signaling, MSCRAMMs, pili-mediated adhesion

## Abstract

*Enterococcus* species are known for their ability to form biofilms, which contributes to their survival in extreme environments and involvement in persistent bacterial infections, especially in the case of multi-drug-resistant strains. This review aims to provide a comprehensive understanding of the mechanisms underlying biofilm formation in clinically important species such as *Enterococcus faecalis* and the less studied but increasingly multi-drug-resistant *Enterococcus faecium*, and explores potential strategies for their eradication. Biofilm formation in *Enterococcus* involves a complex interplay of genes and virulence factors, including gelatinase, cytolysin, Secreted antigen A, pili, microbial surface components that recognize adhesive matrix molecules (MSCRAMMs), and DNA release. Quorum sensing, a process of intercellular communication, mediated by peptide pheromones such as Cob, Ccf, and Cpd, plays a crucial role in coordinating biofilm development by targeting gene expression and regulation. Additionally, the regulation of extracellular DNA (eDNA) release has emerged as a fundamental component in biofilm formation. In *E. faecalis*, the autolysin N-acetylglucosaminidase and proteases such as gelatinase and serin protease are key players in this process, influencing biofilm development and virulence. Targeting eDNA may offer a promising avenue for intervention in biofilm-producing *E. faecalis* infections. Overall, gaining insights into the intricate mechanisms of biofilm formation in *Enterococcus* may provide directions for anti-biofilm therapeutic research, with the purpose of reducing the burden of *Enterococcus*-associated infections.

## 1. Introduction

As an opportunistic pathogen, the *Enterococcus* genus is notorious for its ability to form biofilms. Enterococcal species are common in clinical settings, being responsible for 14.7% of healthcare-associated infections in adults [1]. They are frequently resistant to multiple commonly used antibiotics, including vancomycin [1,2]. Enterococcal biofilms have been detected in a variety of infections, such as periodontal [3], wound [4] and urinary tract infections [5], as well as on medical devices such as ureteral stents [6] and intravascular catheters [7]. While not traditionally considered a respiratory pathogen, *Enterococcus faecalis* has been detected in lower respiratory tract infections [8,9] and has been shown to form biofilms on endotracheal tubes [10,11].

*E. faecium* is another clinically important species that frequently causes healthcare-associated infections, such as urinary tract infections (UTI) and bacteriemia, with frequent multi-drug resistance, and particularly concerning—and increasing—vancomycin resistance [12].

The *Enterococcus* genus also includes a wide variety of species besides *Enterococcus faecalis* and *Enterococcus faecium*, which are gathered in a group called non-faecalis non-faecium *Enterococcus* or Other *Enterococcus* (OE) [13]. The OE group also can be classified into *vanC*-positive OE (including *Enterococcus gallinarum* and *Enterococcus casseliflavus*) and *vanC*-negative OE (the other species). Although the *vanC*-positive species have a high clinical relevance due to the presence of an intrinsic resistance to glycopeptides, via the *vanC* gene, their involvement in biofilm formation has not been extensively studied so far [13]. Also, the *vanC*-negative OE remains a rare encounter in the clinic setting. However, recent studies have highlighted that, for some species like *Enterococcus avium*, *Enterococcus durans* and *Enterococcus raffinosus*, there might be a site-specific association; however, a definitive conclusion has not yet been made and their involvement in biofilm formation has also not been widely described [14].

The process called biofilm formation comprises four stages: Initial attachment, microcolony formation (in part correlated with quorum sensing), biofilm maturation and dispersal. Each stage is governed by specific genes for each species of *Enterococcus*, some being very similar to genes across multiple genera [15]. Numerous morphological factors contribute to their pathogenesis via the production of biofilms. Biofilm-related virulence factors can be divided into secreted factors (cytolysin, secreted antigen A and gelatinase) and cell surface factors, which include pili and microbial surface components that recognize adhesive matrix molecules (MSCRAMMs) and aggregation substances (ASs) [16]. Biofilm formation is regulated by a mechanism called quorum sensing. This is a cell-density-dependent mechanism of intercellular communication, mediated by the expression of specific bacterial genes [15].

This paper aims to compile the existing literature concerning the various factors involved in the formation of biofilms by enterococci and to provide a comprehensive insight into this fascinating niche, while also proposing a number of directions for future research into biofilm eradication techniques that may target these factors. Although there are several articles that review the different steps involved in the formation of biofilms by the *Enterococcus* genus [15,17,18,19,20], the present paper offers a more comprehensive overview of the complexity of factors that govern the process of biofilm formation for this bacterium. It also aims to provide some directions for the future research of anti-biofilm molecules.

We organized the review into four parts: (a) biofilm-related secreted virulence factors; (b) biofilm-related surface proteins; (c) quorum-sensing molecules; and (d) regulators of extracellular DNA release. An overview of the factors discussed is presented in Table 1.

## 2. Biofilm-Related Secreted Virulence Factors

### 2.1. Gelatinase

Gelatinase is a class 2 metalloproteinase that uses zinc, is synthesized by the gene *gelE* in *E. faecalis*, and is composed of 318 amino acids. The main function of this protein is the maturation of the enterocin O16 and the activity of hydrolase against the gelatin, collagen, fibrin, fibrinogen, hemoglobin, complement components C3, C3a, and C5a, endothelin-1, casein and some other small peptides [20,22]. In order for gelatinase to exhibit its protease activity, the 14 C-terminal amino acid needs to be removed. The gelatinase *gelE* gene, part of the *gelE-sprE* operon, together with two other loci (*ef1097* and *fsrABDC*), are regulated by the *fsr* quorum-sensing system. The transcription of *gelE* occurs because of the two-component system: the first component is a membrane-bound histidine kinase (HK) that detects extracellular signs, hazardous substances or other factors modifying the pH, the osmolarity and redox status of the extracellular environment and, in response, auto phosphorylates itself, and the second component is a response regulator (RR), which regulates DNA transcriptions and, therefore, initiates cellular response [46,47,48].

One of the properties of gelatinase which is most relevant to biofilm formation is its ability to degrade the collagen adhesion protein (Ace), which helps the bacteria adhere to other bacteria and to both biotic and abiotic surfaces [49]. By degrading Ace, gelatinase contributes to dissemination and colonization. Together with a serine protease, gelatinase has an important role in N-acetylglucosaminidase (AtlA) regulation. This enzyme is important in forming the extracellular DNA present in biofilms via the degradation of other bacterial cells. Gelatinase functions as a stimulant for AtlA release, while serine protease inhibits this process. AtlA is the main autolysin (out of three, which are AtlA, AtlB and AtlC) involved in biofilm formation and has three domains. The first one is responsible for peptidoglycan adherence, the second one is responsible for the glycosaminidase activity, while the third one if not yet fully understood [50]. Gelatinase also directly causes cell lysis. It can latch onto serine protease, inhibiting it in response, or onto the peptidoglycan, rupturing the cell wall in the process and releasing AtlA into the extracellular matrix [51,52].

### 2.2. Cytolysin

Cytolysin is a protein closely related to lantibiotics, a class of bacteriocins that have lanthionine, methyllanthionine, dehydroalanine and dehydrobutyrine as their main amino acids [24], and that is encoded on the pAD1 sex pheromone plasmid or on the same pathogenicity island as the aggregation substance (AS). Cytolysin comprises two polypeptide chains, a small one called CylLS (63 amino acids) and a larger one named CylLL (68 amino acids) [21]. Separately, these subunits have an important role in the quorum-sensing mechanism via the CylR1–CylR2 complex. The operon responsible for cytolysin synthesis consists of eight genes, out of which two are transcribed divergently (the ones responsible for CylR2–CylR2 synthesis) [23]. The role of cytolysin is to destroy other bacteria, targeting especially Gram-negative bacteria, and eukaryotic cells such as red blood cells. Hemolysis has been observed on solid medium but not on liquid cultures. Cell death is necessary for biofilm formation. When cytolysin is present in the medium and other bacteria start to die out, extracellular DNA is formed, which is essential in biofilm synthesis [21,23,24].

After the transcription and translation of the *cylLL* and *cylLS* genes, they undergo a series of changes both intracellularly and extracellularly. Firstly, CylM (a 993 amino acid polypeptide), synthesized by the 5th gene in this operon (*cylM*), induces changes characteristic of lantibiotics [22]. CylM, also named cytolysin synthetase, has two domains, the first of which catalyzes seven reactions involved in the dehydrogenation of serine and threonine residues, turning them into dehydroalanine and dehydrobutyrine, respectively, and the second of which catalyzes three cyclisation reactions, which include the addition of cysteine residues to dehydroamino acids on the large subunit. Thus, CylM generates CylLL* and CylLS*. Both possess a residue with 16 identical amino acids, which helps in CylB recognition. These two intermediates undergo further changes via *CylB*, the 6th gene in the operon. CylB has two domains [21,23]. The C-terminal domain is an ATP-binding transporter. It takes both subunits and exports them out of the cell. The N-terminal one acts as a proteolytic site, cleaving 24 and 36 amino acids, respectively, from CylLL* and CylLS* outside the cell, thus making CylLL’ and CylLS’, the 2nd intermediate. ATP is needed to transform CylLS*, but not needed to transform CylLL*. CylA, the 7th one in the operon, acts on CylLL’ and CylLS’, cleaving six amino acids (starting from a serine) from the N-terminal end, making CylLL” and CylLS”, two toxic and active subunits. The last gene on the operon, *cylI*, makes a protein CylI, which acts as a defense mechanism against autolysis by interacting with either subunit via an unknown mechanism [21,23,24].

The CylR1–CylR2 complex is a DNA-binding transmembrane complex that induces the synthesis of the cyl operon. It does not resemble other quorum-sensing mechanisms that use a histidine kinase as a means of activation. CylR1 is theorized to act as the transmembrane subunit, while CylR2 interacts with a promoter region located between the *cylR1* and *cylR2* (leftward) genes and the rest of the operon (rightward). The mature CylLS’ is capable of interacting with CylR1, leading to the expression of the operon. This mechanism is of utmost importance when discussing cell proximity. When approaching a Gram-negative bacterium or a eukaryotic cell, the large subunit latches onto the cell, while the small subunit acts as an autoinducer [15,21,23,24,53].

### 2.3. Secreted Antigen A (SagA)

*SagA* is a gene that encodes a major secreted antigen of *Enterococcus faecium*, SagA. It facilitates broad-spectrum extracellular matrix binding (ECM) to fibrinogen, fibronectin, laminin, type-I and type-II and type IV collagen [25]. The deletion of SagA decreases biofilm formation. The SagA protein is a peptidoglycan hydrolase that prefers crosslinked Lys-type peptidoglycan fragments. In *E. faecium*, it is employed in assembling a signature cell wall architecture, which generates smaller muropeptides that more effectively activate nucleotide-binding oligomerization domain-containing protein 2 (NOD2) in mammalian cells [15,54].

## 3. Biofilm-Related Surface Proteins

### 3.1. Pili

Pili extend from the surface of bacterial cells and are involved in a variety of functions, such as motility, adhesion, and conjugation. In *E. faecalis* and *E. faecium*, pili play an important role in the attachment of the bacteria to host cells and in the formation of biofilms, which can protect the bacteria from the host’s immune system and antibiotics. The specific characteristics and functions of bacterial pili can vary depending on the bacterial species and the type [55].

In *E. faecalis*, the Ebp (endocarditis and biofilm-associated pili) proteins, namely EbpA, EbpB, and EbpC, are components of Ebp pili; these have been shown to be essential for biofilm formation in vitro with regard to endocarditis in rats [26], as well as urinary tract infections in mice [27].

There are two different types of pili at the surface of a single *E. faecium* bacterium, pilA and pilB, which both contribute to biofilm formation [28]. They are phenotypically distinct, due to the fact that they are composed of different pilin subunits. Bacteria usually need to be in a high-temperature environment in order to form pili. The standard temperature range in hospitals is 21 °C to 24 °C, which is too low for bacteria to form pili; however, the pilin subunits are shown near the membrane [56,57].

PilB are thicker, more flexible, and they are expressed at the poles of the dividing cells. PilB are often involved in conjugation, and they are expressed in the growth phase. During cell division, the major pilin subunits for pilA and pilB are formed. PilB pilin is translocated around the cross wall (the newly synthesized peptidoglycan layer between the two new cells) and deposited in murein sacculi during its separation. When the cell grows, new murein sacculi arise and the old ones are pushed to the poles, with pilB proteins inside [15,56,57].

There are three stages of bacterial growth: early exponential (EE), exponential (E) and stationary (S). The expression of the PilB-type pili in dividing cells is different in each of these stages. As the cell grows, so does the level of pilin protein. Later on, in the exponential phase, the murein sacculi approach the plasma membrane and form PilB. The newly formed pilin subunits are deposited in the murein sacculi near the cross wall. PilB pili are thicker than pilA, which indicates the incorporation of more pilin subunits [55]. This process is illustrated in Figure 1.

The other type of pili in *E. faecium* is PilA-type pili. These types of pili are absent in cell division. However, the *pilA* gene and the pilA pilin are present throughout the cell cycle. Their role is to adhere the bacteria to a substrate or to another cell. On liquid media, PilA cannot grow, therefore solid media are used to observe these types of structure [33,58].

Given that pili have been shown to contribute to the formation of biofilms by *E. faecalis* [26,59] and *E. faecium* [28], it would be reasonable to assume that the inhibition of pili formation would inhibit enterococcal biofilm formation. This assumption is further sustained by the fact that this phenomenon has been observed in other bacteria such as *Pseudomonas aeruginosa* [60]. Potential targets for inhibiting biofilm formation could include genes involved in pili formation, such as *Cpd* and *Ccf* for *E. faecalis*, and *pilA* for *E. faecium* [28,36]. Another way to eradicate the multi-resistant bacteria that form biofilms could be the disruption of pili-mediated attachment, targeting pilA. In *E. faecalis*, blocking Ebp using an anti-Ebp vaccine has been shown to inhibit biofilm formation in a mouse CAUTI (catheter-associated urinary tract infection) model [61].

### 3.2. Microbial Surface Components Recognizing Adhesive Matrix Molecules (MSCRAMMs)

MSCRAMMs are a family of bacterial adhesins capable of binding to the elements of the extracellular matrix, predominantly to collagen. This family comprises three proteins: Ace, Acm and Scm. Ace is found in *E. faecalis*, while Acm and Scm are found in *E. faecium*. These surface proteins are usually characterized by an amino acid sequence of leucine, proline, x (any other amino acid) and glycine [30,33].

#### 3.2.1. Ace

The *ace* gene (adhesion to collagen of *E. faecalis*) encodes an adhesin that binds to collagen (types I and IV) and laminin. Ace is a CNA-like MSCRAMM that binds to collagen via the multistep ligand-binding mechanism called the collagen hug [29,30]. Most proteins in the CNA-like MSCRAMM sub-family have been shown to act as virulence factors in experimental bacterial infections [62].

Ace expression is dependent on external factors such as the presence of bile salts and serum, as well as internal factors such as the putative Ers box, a nucleotide sequence (AACATTTGTTG) encoding a protein that, in turn, acts as a positive regulator of virulence genes; it also regulates *GelE* expression, which, in turn, is dependent on a complete *fsr* system [49,63]. Studies have shown that the administration of monoclonal anti-Ace antibodies (mAb), specifically mAb70, reduces *E. faecalis* infections in rat models [64].

#### 3.2.2. Acm

In *E. faecium*, the *acm* gene (adhesion to collagen of *E. faecium*) has been identified and found to encode the Acm protein. Acm presents a structure with a N-terminal signal peptide, followed by an A domain, various B sequences (depending on the species) and a C-terminal sequence that enables anchoring to the peptidoglycan layer. In previous studies, considerable homology has been demonstrated with other collagen-binding adhesins present in *Staphylococcus aureus* and *E. faecalis*. Like Cna in *Staphylococcus aureus* (the prototype for the adhesin family), the ligand-binding A domain has three subunits, N1, N2 and N3. The N1N2 subdomains bind with affinity to collagen, through a mechanism called the collagen hug mechanism [31,32].

As has been already demonstrated in samples derived from clinical isolates, there is a correlation between the adherence levels of the pathogen and the amount of Acm protein expressed on the cell surface. The expression of *Acm* by *E. faecium* and its adherence to collagen appear to be important parameters for virulent expression in patients with endocarditis, as shown by Nallapareddy et al. [31]. As explained by the collagen hug mechanism and by the reactivity of the serum to the recombinant form, the use of specific antibodies against Acm would facilitate the inhibition of collagen adherence and is therefore a promising therapeutic and prophylactic strategy in patients at risk of *E. faecium* infections [15,32].

Another important element involved in the pathogenesis of *E. faecium* producing biofilm infections is certainly autolysin. Autolysin is an endogenous lytic enzyme that is present in all peptidoglycan-containing bacteria. In addition to the breakdown of the peptidoglycan component of cells, it has an important function in cell separation during division and in the release of extracellular DNA (eDNA) into the biofilm matrix, ensuring its stability [50,53]. In *E. faecium*, the major autolysin identified is AtlAEfm, which appears to be involved in the localization of Acm on the bacterial cell surface, contributing to collagen adherence and the pathogenesis of infections. Therefore, given the fundamental functions expressed in the pathogenic process, as concluded for Acm, AtlAEfm represents a potential target in the treatment of *E. faecium* producing biofilm infections [31].

#### 3.2.3. Scm

Due to the increase in nosocomial infections caused by *E. faecium,* more studies have had this bacterium as their main focus. In a recent article, a genomic analysis of the TX0016 strain of *E. faecium* has shown the importance of Scm (second collagen adhesin of *E. faecium*, previously known as Fms10 from cell-wall-anchored *E. faecium* surface protein), a MSCRAMM [33].

In addition to Acm, which binds mostly to type I collagen, the main component of collagen fibers in human tissue, Scm is an adhesin with a much greater affinity for collagen type V, a key component of the cross-linkage between interstitial collagen fibrils and membranous collagen networks. Having two types of collagen adhesins gives *E. faecium* the ability to fine-tune its adherence phenotype to better suit the given tissue. For collagen type V, which is abundant in the intestinal submucosa, Scm might be the key factor that causes its resistance and persistence in the GI tract [65].

### 3.3. Aggregation Substance (AS)

The aggregation substance is a hair-like glycoprotein on the bacterial surface. It is synthesized on “older” structures of the *Enterococcus* cell wall from the *asa1* gene, located on a sex pheromone plasmid, pAD1, when exposed to a sex pheromone, cAD1 [34]. Asc10, also called the aggregation factor, is encoded on the pCF10 plasmid, and it is expressed while within the mammalian bloodstream. Its main role is to help bacteria adhere to other bacteria and eukaryotic cells, and to diminish superoxide function [66,67]. The details of this mechanism are schematically depicted in Figure 2.

Out of the 1296 amino acids, the main two parts on the AS are the two RGD motifs (rich in arginine, glycine and aspartate). It also has a signal domain comprising 43 amino acids and a C-terminal proline-rich sequence, which is important regarding integration into the bacterial cell wall and membrane. RGD is a sequence comprising three amino acids that is composed of arginine, glycine and aspartic acid, which helps in adhesion. This motif interacts with the β subunit of the integrins of eukaryotic cells, mainly macrophages and epithelial cells, as well as the enterococcus-binding substance (EBS), which is similar in structure to lipoteichoic acid. The N-terminal part of the protein is the one that interacts with both integrins and enterococcal surface proteins, while the C-terminal region is bound to the cell wall. Experimental studies have shown that when N-terminal amino acids are eliminated, biofilm formation decreases significantly compared to the C-terminal mutant counterparts [66,68].

AS also helps with bacterial aggregation. The F- bacterium secretes small peptides called sex pheromones. When the F+ bacterium senses these pheromones, it starts to transcribe the asa1 gene and express AS on the cell surface. When AS interacts with EBS, conjugation occurs, after which both bacteria possess the pAD1 plasmid [68]. The expression of AS leads to bacterial clumping and higher antibiotic resistance, helping in biofilm formation. Because of the aggregation, sonication is needed in order to study a single bacterium. Besides its aggregating properties, AS also helps with regard to integration and resistance inside the macrophages (MF). Lucigenin was used in order to detect superoxide activity inside the MF, and it was shown that CL (a metabolite of lucigenin) had a lower concentration in asa1+ cells in the first 3 h after integration, according to Süßmuth et al. [34]. This indicates that AS lowers superoxide activity and thus helps *E. faecalis* survive for longer periods of time inside the cell.

The RGD motifs are recognized by the beta subunits of integrins. The main ones are CD18 and CD11b on macrophages, helping with bacterial integration. The same region (RGD) helps in extracellular matrix binding. The presence of AS on the surface of the enterococci increases the adherence to both fibronectin (8-fold) and thrombospondin (4-fold). Monoclonal antibodies against CD18 and CD11b have been shown to reduce bacterial integration, prohibiting bacterial dissemination throughout the organism. RGDS (RGD serine)-containing peptides latch onto the integrins and EBS better than RGD, thus entering into competition with AS. This way, bacteria cannot interact with the surface or with other bacteria, thus reducing the risk of biofilm formation [34,66,68,69].

## 4. Quorum-Sensing Molecules

The quorum-sensing process works through the synthesis and release of small signal molecules called pheromones. Pheromones are small signal molecules that stimulate an intercellular response among members of the bacterial community. Both Gram-negative and Gram-positive bacteria use this type of cell-to-cell communication [70]. In Gram-positive species, this communication consists of small peptides called autoinducers or quorum-sensing peptides, which are very sensitive to their own receptors. These molecules express an important function in communication between microbes, influencing biofilm stability and development, but are also possibly involved in communication with the host environment [70]. In *Enterococcus* spp., biofilm formation is regulated by the *fsr* (fecal streptococci regulator) locus and peptide pheromones. In addition to the communication between microbes, pheromones appear to mediate the transfer of plasmids through the conjugative apparatus. This process facilitates the transfer of genes that promote biofilm formation, such as adhesins involved in the pathogenesis of the infection. Some examples of pheromones in the enterococcal species are Cpd, Cob and Ccf. As demonstrated by Eaton et al. in vitro, these pheromones are chemotactic for human leukocytes, being able to induce the production of superoxides and the secretion of lysosomal enzymes. Therefore, they can be considered virulence factors [71].

These peptide pheromones consist of sequences with seven to nine amino acids and are synthesized by ribosomes. After that, they undergo post-translation modifications and become activated during excretion. Secretion is mediated by a membrane-associated ATP-binding cassette (ABC), which facilitates the achievement of a concentration threshold; this activates a specific receptor through phosphorylation and consequently the transcription of the target gene [70,72]. Details of these mechanisms are explained in Figure 3, Figure 4 and Figure 5.

### 4.1. Cob1

Cob1 is a sex pheromone isolated from *E. faecalis* that consists of eight hydrophobic amino acids, H–Val–Ala–Val–Leu–Gly–Ala–OH, and induces the conjugal transfer of hemolysin and bacteriocin plasmids (pOB1 and pYI1). Besides these plasmids, Cob1 influences the relationship with the host environment, through the expression of a surface adhesin that mediates the aggregation between recipient and donor cells (aggregation substance), and also the sexual aggregation between donors; it is therefore called a “clumping-inducing agent” (CIA) [35].

In *E. faecalis*, the transfer of the pCF10 plasmid from donor cells is mediated by the secretion of the cCF10 pheromone from the recipient cells. Donor cells, containing pCF10, internalize the cCF10 pheromone and induce the expression of the aggregation substance (Asc10), which mediates plasmid transfer. This paracrine pheromone signaling pathway is the link between sex pheromone secretion and biofilm formation [66,70,73].

As demonstrated in previous studies, there is a mechanism to prevent the autocrine activation of this process: iCF10 and PrgY. The former is a short protein that acts as a competitive signaling inhibitor on the pheromone, while PrgY directly degrades the CF10 pheromone. The competitive signaling inhibitor iCF10 has been shown to be degraded by a blood plasma component, inducing activation of the conjugation system and thereby increasing the degree of colonization in the host [74,75]. The details of this process are explained in Figure 6. Understanding the synthesis and expression of the pheromones signaling pathway in biofilm-producing bacteria is fundamental in the approach to determining the new therapeutic targets of multi-drug-resistant (MDR) bacteria.

In the case of commensal *E. faecalis*, cOB1 is encoded by a signaling lipoprotein, EF2496, which possesses the precursor sequence of cOB1. Through a zinc-dependent metalloprotease-mediated cleavage, cOB1 is transported out of the cell by a ATP-binding transporter named the peptide pheromone transporter (PptAB). This proteolytic cleavage of the lipoprotein signal peptide shows similarities to Amyloid β formation in Alzheimer’s disease, and it is a critical step in the production and interaction of cOB1 [76]. As demonstrated by Gour et al., the monomeric form of cOB1 is capable of binding to the specific receptor and of inducing the conjugative transfer of plasmids, while the aggregated form does not bind to the receptor, blocking gene transfer [76]. This inhibition could be a functional therapeutic target against the propagation of virulence factors mediated by peptide pheromones.

As demonstrated by Gilmore et al., when the cOB1 pheromone is released and comes into contact with its receptor on the MDR *Enterococcus* V583, it modulates the expression of the genes [77]. The major effect of the cOB1 expressed by the commensal species on V583 is to induce transcription of the pTEF2 genes and insertion-sequence-like element accumulation on the V583 chromosome. This interaction results in an incompatibility between V583 and the fecal consortium, causing the killing of V583. Therefore, although little is known about the colonization mechanisms, the destabilization of the native gastrointestinal flora facilitates the colonization of species such as MDRV583 *Enterococcus*.

### 4.2. Ccf

The *CcfA* gene is responsible for the production of the cCF10 pheromone. The CCF10 pheromone has a molecular weight of 789 and has the following amino acid structure: H–Leu–Val–Thr–Leu–Val–Phe–Val–OH [67]. It is necessary for the transfer of the pCF10 tetracycline-resistant plasmid in vivo. The transfer of pCF10 is mediated by two different peptides: one inhibitor, iCF10, and one pheromone, cCF10, both expressed by important genes in *E. faecalis*. They both interact with the PrgX receptor protein. The transformation of the cCF10 precursor in mature pheromone cCF10 is mediated by another gene called eep, which separates the amino acid terminal end form the rest of the protein [78]. The result is a more hydrophobic amino acid.

The production of the cCF10 pheromone by *E. faecalis* can be increased by introducing a cloned *ccfA* gene. It can also make non-pheromone producers, for example, *Lactococcus lactis,* to produce cCF10 pheromones. Besides plasmid transfer, the cCF10 pheromone can also induce aggregation [36].

### 4.3. Cpd

The *cpd* gene is responsible for the production of the cPD1 pheromone, which mediates the transfer of the pPD1 plasmid. A cell synthesizes the pPD1 plasmid and donates it to another. This plasmid is responsible for synthesizing proteins involved in the mating response to the cPD1 sex pheromone, which binds to the TraA intercellular receptor. This transfer can be blocked by the iPD1 inhibitor, produced by the recipient cells, which is a competitive inhibitor that binds with TraA before the pheromone [37,38,79]. TraC is a pheromone-binding protein, encoded by the pPD1 plasmid, that enables recipient cells to receive cPD1. The cPD1 pheromone has a higher affinity for TraA than for TraC. However, TraC plays an important role in the crossing of the cell wall by the pheromone, acting like a protein carrier. PrgZ, the protein that binds with cCF10, is very similar to TraC. Because of this similarity, cCF10 can also bind with TraC. However, prgZ, encoded by the pCF10 plasmid, has no affinity for cPD1 [35,37,38,79]. The cPD1 pheromone has a molecular weight of 912 and has the following aminoacidic structure: H–Phe–Leu–Val–Met–Phe–Leu–Ser–Gly–OH [79]. The details of this mechanism are presented in Figure 7.

### 4.4. Eep

A gene known to be involved in quorum sensing is *eep*. It is involved in the maturation of the peptide pheromones cAD1 and cCF10 through a membrane protease Eep. The protease encoded by *eep* comprises 422 amino acids and uses Zn2+ as a cofactor. It also facilitates the proteolytic processing of RsiV, the anti-sigma factor for sigV, which is an extracytoplasmic function (ECF) sigma factor, improving stress resistance [39,80].

The innate ability of *E. faecalis* to resist lysozyme stress is offered by the complete degradation of RsiV, which is a counterpoint to the activation of SigV. In eep deletion, mutant strains were shown to only have a partially degraded RsiV. Eep works together with AhrC and the ArgR, arginine repressors that block the uptake of this amino acid or activate its catabolism, leading to biofilm formation. The deletion of genes that encode these proteins has been shown to lower the burden in UTI and endocarditis [39,80,81].

### 4.5. The fsr Operon and ef1097 Locus

This system is composed of four genes: *fsrA* (RR), *fsrB*, *fsrD* and *fsrC* (HK), in order in the *fsr* operon. When *fsrC* is transcribed, a membrane-bound HK—*fsrC*—results, with the role of sensing an 11-amino-acid-long cyclized peptide lactone in the extracellular environment in the form of gelatinase biosynthesis-activating pheromone (GBAP). When the *fsrD* gene is transcribed, a propeptide *fsrD* is formed, which undergoes further changes mediated by *fsrB*. fsrB is a transmembrane protein that belongs to the accessory regulator protein class (AgrB-accessory gene regulator). This protein modifies *fsrD*, turning it into GBAP, which will further interact with *fsrC* [44,45,82]. This interaction will lead to *fsrA* (RR) activation (phosphorylation), a protein belonging to the lytTR family of DNA-binding domains, which will in turn act as a facilitator for the transcription of the *ef1097* locus (composed of two genes, *ef1097* and *ed1097b*), *fsrB* and *gelE,* facilitating *ef1097*, *ef1097b*, the *fsr* locus, *gelE* (encoding a gelatinase), and *sprE* (encoding a serine protease). If the *fsr* quorum-sensing system is affected, these genes, together with 75 others, are inhibited, including a gene important in biofilm formation, *bopD* [42].

The role of the *ef1097* locus is the synthesis of enterocin O16, an antimicrobial peptide (bacteriocin) composed of 68 C-terminal amino acids [22]. This peptide inhibits the proliferation of *Lactobacillus* spp., but it does not affect *Staphylococcus*, *Enterococcus* or *Listeria* growth; at higher concentrations, it can also act as an antifungal peptide. This peptide is obtained after EF_1097 (the precursor), a 191-amino-acid-long peptide, is delivered out of the cell via the Sec system, which cuts the peptide between Ala56 and Ser57, and when gelatinase cuts the precursor at the 123rd amino acid, leaving a 68-amino-acid-long peptide residue, previously named enterocin O16 [22]. Unlike in *Streptococcus dysgalactiae*, the genes associated with bacteriocin resistance (*dysI* to *dysA*, the gene that codes for dysgalacticin, in *S. dysgalactiae*) are not located on the same operon. It has been shown that *ef1097b* does not confer resistance to enterocin O16 as was expected [22,44,45,82].

## 5. Regulators of Extracellular DNA Release (eDNA)

The function of eDNA as a component of the biofilm matrix is well established [43,83]. eDNA is known to perform various biological functions in bacteria (adhesion, early biofilm development, stabilization of the biofilm matrix, horizontal gene transfer, phagocytosis prevention, inhibition of inflammation) that vary with the different phases of the biological cycle [43,50]. However, recent studies have remarked upon the lack of a direct correlation between microbial activity and the iDNA/eDNA ratio, as well as the limitations of the current extraction method used for qPCR. The mechanisms that regulate eDNA release from bacterial cells are also less clear. At first, eDNA was only thought to derive from cell lysis, while recently, numerous studies have demonstrated that the release of eDNA in prokaryotic and eukaryotic cells can result from lysis-dependent and lysis-independent mechanisms. This being stated, there is an opportunity for further studies concerning eDNA release in particular species [15,43,83].

In Gram-positive bacteria, endogenous autolysin (*AtlA*) stimulates eDNA release via QS-dependent lysis. In *E. faecalis*, eDNA production is QS-dependent, and the lactone peptide (*FsrD*) triggers the activation of two proteases: gelatinase (*GelE*) and serine protease (*SprE*). These two proteases, through different mechanisms, induce the release of eDNA [20,43,83].

*E. faecalis* produces several autolysins, the most frequently identified being AtlA. This is N-acetylglucosaminidase, an endogenous lytic enzyme that is crucial in the separation of daughter cells during cell division. *AtlA* sequencing showed a structure composed of three domains: domain I, which hash unknown functions, domain II, which contains the region capable of hydrolyzing peptidoglycans, and domain III, which is composed of six LysM residues recognizing the N-acetylglucosamine residues (GlcNAc) of cell wall peptidoglycans, as is required for AtlA anchoring. Some studies have demonstrated that the inactivation of AtlA results in increased chaining, due to defect septum cleavage, decreased primary attachment, and decreased biofilm production. Consequently, the fundamental role of AtlA can be deduced not only in cell growth and lysis, but also in the initial adhesion and eDNA release in the biofilm composition during the accumulation phase in *E. faecalis* [20,22,43].

As mentioned previously, QS-dependent eDNA production in *E. faecalis* is also mediated by the activation of two extracellular proteases. The gene coding for gelatinase (*GelE*) is located adjacent to *fsrC* and is co-transcribed with serine protease (*SprE*). Mutations in the *fsr* locus, resulting in *GelE* defects, negatively affect its ability to produce biofilms, indicating an important role in the activation of autolysin and in the regulation of the biological mechanisms of biofilm production. As reported by Thomas et al., there are two possible mechanisms through which these proteases may express the regulatory effect of *Atla* and influence biofilm development [50,84].

The first mechanism, defined as the autolytic pathway, is that in which *GelE* localizes itself on the cell wall of producing cells in order to activate autolysis. The levels of *SprE* present could counteract *GelE*-induced autolysis but, if insufficient, the cell undergoes autolysis.

The second mechanism, defined as allolytic or fratricidal, involves the diffusion of *GelE* from a producer cell to a target sister, so as to induce the autolysis of the sister cell. Both types of autolysis, activated by *GelE*, involve the mediation of *AtlA*, making these theoretical models similar to the process of programmed cell death in eukaryotic cells. The cytotoxic activity expressed by *GelE* (the effector) towards a producer cell (autolysis) or sister cells (allolysis) through the mediation of *AtlA* would result in the release of *eDNA*, essential in biofilm development. The allolysis (sibling killing sibling) model considers the concept of cell developmental competence, where the inability of a subpopulation of cells to produce *SprE* (the regulator, immune factor) would result in the death of these cells [43,44,45].

Previous studies have demonstrated the possibility that, following the attachment phase, the *fsr* quorum-sensing signaling pathway could be activated, resulting in the expression of *GelE*, *SprE* and *AtlA*. When *GelE* reaches high levels of activity in biofilm production, at later stages of colonization, *AtlA* could induce cleavage, resulting in small chain cells of *E. faecalis*. This configuration would increase microbial dissemination and evasion from the host’s defense. In fact, some host immune components, such as LL37, defensin and complement components (C3a and C3b), appear to be able to undergo the proteolytic activity of gelatinase. In addition, gelatinase has been shown to adhere to fibrin, increasing dissemination and providing further evasion of the host immune system. All these data demonstrate how the *GelE*-mediated autolytic mechanism affects the pathogenic process and the virulence of *E. faecalis* producing biofilm [43,44,45,52].

Barnes et al., in order to confirm the primary role of eDNA in biofilms, compared the levels of extracellular DNA between samples of *E. faecalis* producing biofilm and a planktonic form, revealing a significantly higher amount in the biofilm samples [85]. The study also enabled a temporal and morphological analysis of eDNA release in the biofilm ecosystem. In the first 4 h post attachment, there is a significant elevation of eDNA release, which remains stable for the following 24 h. From a morphological point of view, eDNA mainly has two structural forms: a long and intercellular complex (yarn structure), and a globular form (sweater structure), which resembles the appearance of the biofilm matrix. Despite these promising results, the mechanisms that induce DNA export and articulation in biofilm matrix stabilization in *E. faecalis* remain incompletely understood. Certainly, as reported by Yu et al., extracellular DNA is a potential therapeutic target in the treatment of biofilm-producing *E. faecalis* infections. In fact, it has been demonstrated that eDNA inhibition decreases its ability to produce biofilms and increases the susceptibility of *E. faecalis* to NaOCl, even at low concentrations (0.5%) [83].

## 6. Conclusions

This paper provides an extensive overview of the factors contributing to biofilm formation in *Enterococcus species*, especially in those involved in human pathology, such as *E. faecalis* and *E. faecium*. Some of the factors we identified in this literature review are secreted virulence factors (gelatinase, cytolysin, sagA), some are surface proteins (Acm, Scm, Aggregation substance), some are quorum-sensing molecules (Cob, Ccf, and Cpd, Eep, the *fsr* operon) and some are molecules regulating eDNA release through cell lysis (AtlA, gelatinase, cytolysin, serin protease). All these factors are involved in biofilm formation, through adhesion, aggregation, matrix formation and stabilization, and some also contribute to gene transfer and bacterial resistance to antibiotics and the immune response.

While some factors are shared between species (aggregation substance), some factors, or, at least, their variants, are species-specific. Even if the overall structure of biofilms, their stages of development and their biological functions are known to be similar across the genus and even multiple genera, there are gaps in the research pertaining to the mechanism of biofilm formation for each particular species. New therapeutic strategies against biofilm-associated infections caused by *Enterococcus* and other multi-drug resistant bacteria can only be developed by understanding these processes and targeting relevant specific genes or molecules.

## Figures and Tables

**Figure 1 ijms-24-11577-f001:**
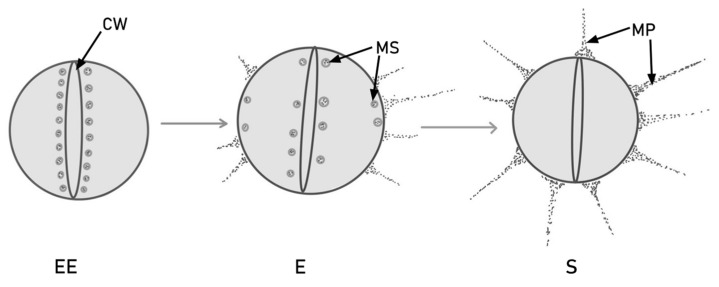
PilB production and translocation via murein sacculi (MS) during cell division, in different stages of colony growth. In the early exponential (EE) phase, the pilin subunits are stored near the bacterial cross wall (CW). Later on, in the exponential (E) phase, some subunits migrate near the extremities and form pili, but others remain around the cross wall. In the stationary (S) phase, all the pilin subunits are converted into mature PilB-type pili (MP).

**Figure 2 ijms-24-11577-f002:**
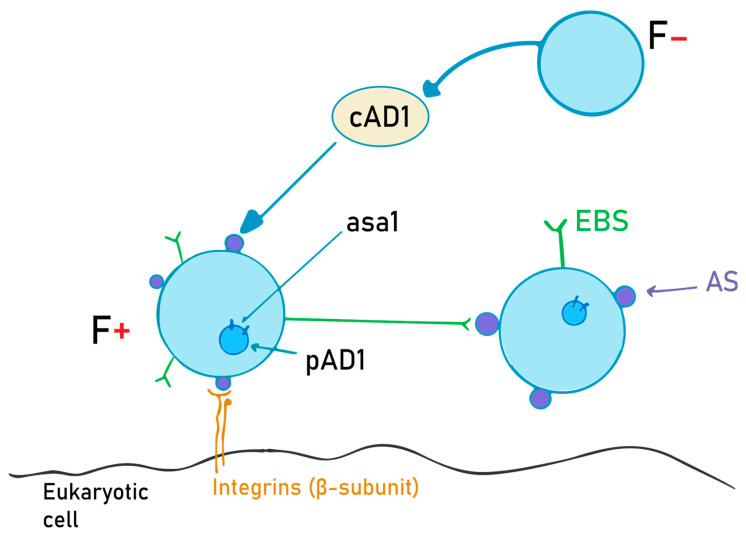
AS (aggregation substance) production and the role of the cAD1 pheromone. F- bacteria emit cAD1, a sex pheromone, in order to induce AS synthesis in F+ bacteria. Once the AS is expressed on the cell surface, it will interact with the EBS (enterococcal binding substance), and thus the two cells can now exchange the pAD1 plasmid (with the *asa1* gene that encodes for AS).

**Figure 3 ijms-24-11577-f003:**
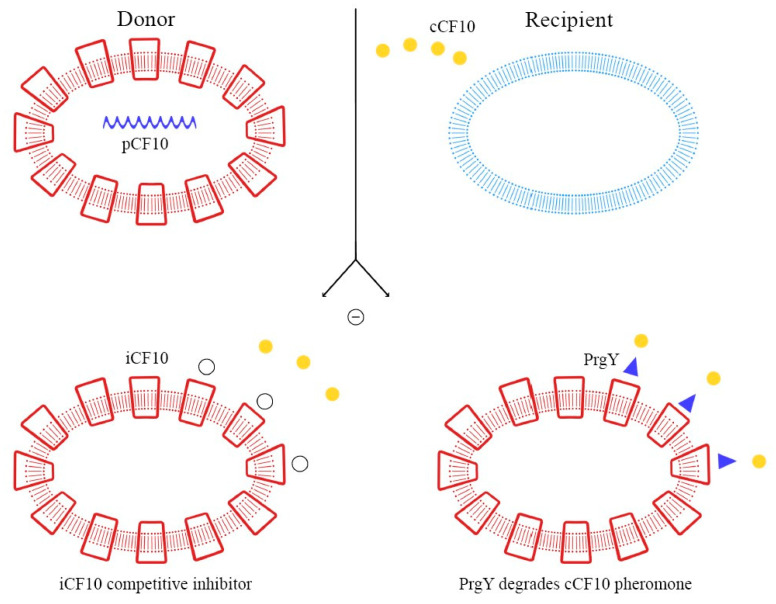
Autocrine inhibition of pheromone-mediated transfer mechanisms.

**Figure 4 ijms-24-11577-f004:**
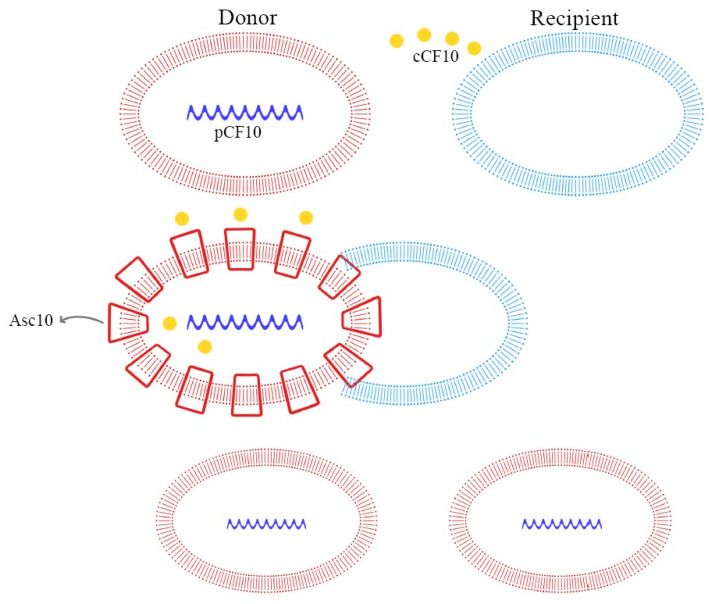
Pheromone-mediated (pCF10) horizontal plasmid transfer and Asc10 expression.

**Figure 5 ijms-24-11577-f005:**
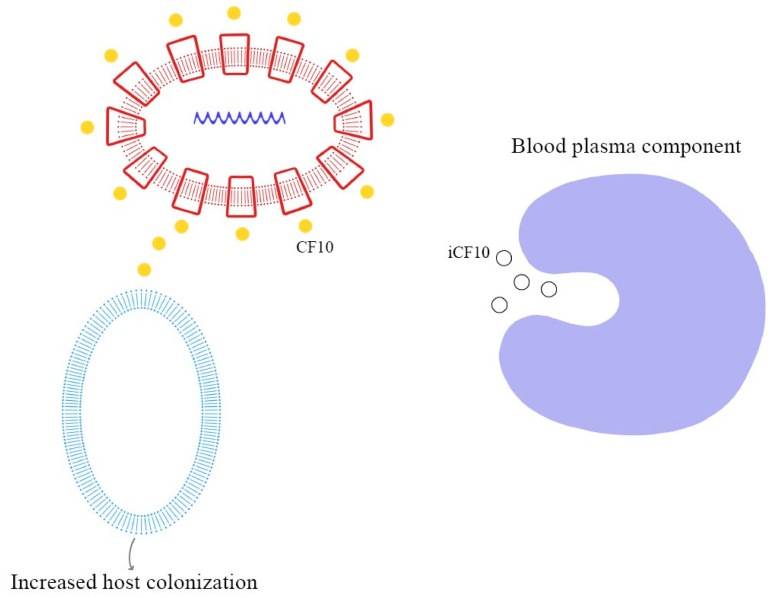
Degradation of iCF10 and activation of conjugation system resulting in host colonization.

**Figure 6 ijms-24-11577-f006:**
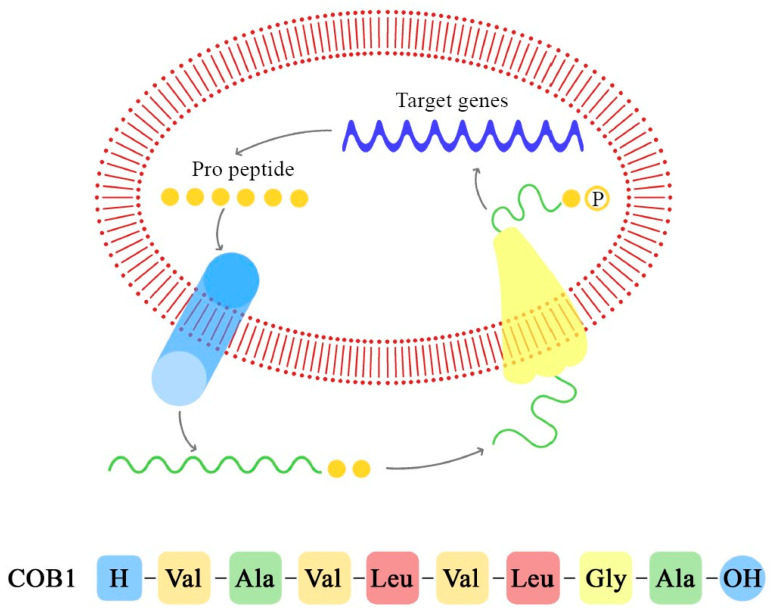
Pheromone synthesis and target gene activation produced by COB1 pheromone (the amino acids sequence is presented below).

**Figure 7 ijms-24-11577-f007:**
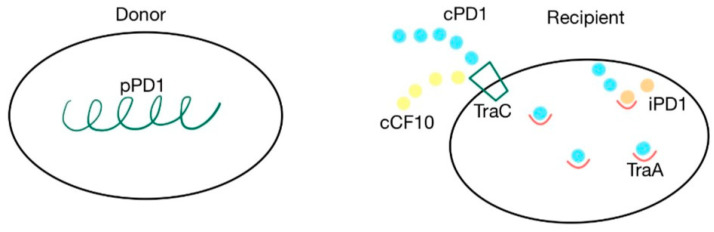
cPD1 pheromone and pPD1 plasmid interaction via TraA receptor. Pheromone-mediated transfer mechanisms between a donor that contains the pPD1 plasmid and the recipient’s TraC and TraA proteins. cPD1 and cCF10 pheromones can both bind with TraC.

**Table 1 ijms-24-11577-t001:** Overview of the factors involved in the production of biofilms by *Enterococcus*.

Category	Factor	Functions	Reference
Biofilm-related secreted virulence factors	Gelatinase	Adhesion; dissemination; eDNA release; enterocin O16 maturation; hydrolysis of gelatin, collagen, fibrin, fibrinogen, hemoglobin, complement components C3, C3a, and C5a, endothelin-1, casein and some other small peptides	[21,22]
	Cytolysin	Bacteriocin; eDNA release	[21,23,24]
	Secreted antigen A (SagA)	Adhesion (broad-spectrum extracellular matrix binding)	[25]
Biofilm-related surface proteins	Ebp	Adhesion; aggregation	[26,27]
	pilA	Adhesion; aggregation	[28]
	pilB	Adhesion; aggregation	[28]
	Ace	Adhesion	[29,30]
	Acm	Adhesion	[31,32]
	Scm	Adhesion	[33]
	Aggregation substance	Adhesion, aggregation, and antibiotic resistance	[34]
Quorum-sensing molecules	Cob1	Aggregation	[35]
	Ccf	Aggregation; bacterial conjugation	[36]
	Cpd	Mediates bacterial conjugation	[37,38]
	Eep	Bacterial conjugation; cellular distribution	[39,40,41]
	*fsr* operon	Facilitates transcription of 76 genes involved in biofilm formation, including BopD	[22]
	*ef1097* locus	Enterocin O16 synthesis	[22]
	BopD	Putative sugar-binding transcriptional regulator	[42]
Regulators of extracellular DNA release	AtlA	QS-dependent lysis	[20,22,43]
	Gelatinase	Autolysis; allolysis	[43,44,45]
	Serine protease	Autolysis; allolysis	[43,44,45]
	Cytolysin	Allolysis	[29,30,31]

## Data Availability

No new data were created or analyzed in this study. Data sharing is not applicable to this article.

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
