# Peer review of "An Overview of the Factors Involved in Biofilm Production by the Enterococcus Genus"

_ijms, 2023, doi:10.3390/ijms241411577_

Round 1

Reviewer 1 Report

To my mind, there is a disagreement between the title and the aim, as outlined at the Lines 16-17. Whereas the title announces a review of new data on the biofilm production, the provided aim declares providing “comprehensive understanding of the mechanisms underlying biofilm formation” and exploration of “potential strategies for eradicating” them. Since there is seemingly no distinction between what is well established in the field of Enterococcus biofilms and what should be considered new, I recommend to shorten the title to “An overview of the factors involved in biofilm production by Enterococcus genus”. Nobody expects to find a recent review based just on old literature.

Section 4. Remove information on Candida albicans. It is a sexual eukaryote, and neither conceptual coverage nor the degree of information synthesis in the paper make place for such distant analogies.

The conclusion should be shortened to one or two paragraphs. There may be an option to transform the current conclusion into a table, summarizing the known factors of biofilm production. The table can be placed in the introduction.

There are several examples of what may seem as common rather than scientific language. Please revise or explain:

Line 14. Species are famous.

Line 15. “Resistance of infections”. Resistance is a property of a bacterium.

Line 17. “Resilient structures”. Check the meaning in a dictionary.

Line 79. “kinase … which sensed external stimuli”. Provide more mechanistic description.

Line 190. “A good way to eradicate biofilms could be inhibition of pili formation”. How good is it? Are there comparative data? Specify the inferior approaches or revise.

Other considerations:

Lines 18-19. “Species-specific genes and virulence factors”. Are the gelatinase and cytolysin genes species-specific? Probably, particular gene variants are species-specific.

Line 21. What is a difference between regulating gene expression and target gene activation?

Lines 28, 32, 33. Italics in the genus name.

Line 48. is mainly comprised of --> comprises

Line 56. Biofilm formation is mediated by quorum sensing. It is probably regulated, not mediated.

Lines 56-57. Check the multiple use of “regulated” and “mediated”.

Lines 62-63. Please cite those articles after the expression “the Enterococcus genus”.

Line 76. Write amino acids in full. Please explain the appearance of sprE.

Line 81. In the introduction, the roles of gelatinase were not mentioned.

Line 97. Change pathogeny to pathogenicity.

Lines 145, 146. The old section title was not deleted.

Figure 1 carries no information related to the main text and should be deleted with the respective note on the Line 145.

Figure 2. Mark phases on the image, and annotate the shown structures, including the elongated O-like structure in the middle of the cells.

There is a need to correct English.

Author Response

Thank you for your extensive and thorough review! We have made the necessary adjustments and we would like to present you with our replies for each of the suggested modifications:

To my mind, there is a disagreement between the title and the aim, as outlined at the Lines 16-17. Whereas the title announces a review of new data on the biofilm production, the provided aim declares providing “comprehensive understanding of the mechanisms underlying biofilm formation” and exploration of “potential strategies for eradicating” them. Since there is seemingly no distinction between what is well established in the field of Enterococcus biofilms and what should be considered new, I recommend to shorten the title to “An overview of the factors involved in biofilm production by Enterococcus genus”. Nobody expects to find a recent review based just on old literature.

R: We aggree with the suggestion, and we have shortened the title accordingly.

Section 4. Remove information on Candida albicans. It is a sexual eukaryote, and neither conceptual coverage nor the degree of information synthesis in the paper make place for such distant analogies.

R: We aggree that the Candida analogy is distant, and we have removed any mention of it.

The conclusion should be shortened to one or two paragraphs. There may be an option to transform the current conclusion into a table, summarizing the known factors of biofilm production. The table can be placed in the introduction.

R: We shortened the conclusion as suggested. We have now created a table in the Introduction section, summarizing the various factors, the species in which they were detected, and the identified functions, with references.

  1. “Species are famous”, “Resistance of infections” – we’ve decided to rephrase the whole sentence using more appropriate language.
  2. “Resilient structures”. Check the meaning in the dictionary. – we checked the meaning of the word “structure” in the Merriam-Webster dictionary and one of the definitions stated “the aggregate of elements of an entity in their relationships to each other”. Since a biofilm represents exactly an aggregation of multiple communities of microbial cells in an extracellular polymeric substance, we consider that the word “structure” accurately represents the meaning of the word “biofilm” from a biological standpoint. The word “resilient”, as per the same dictionary, means the capacity “of withstanding shock without permanent deformation or rupture” which, again, is a key feature of the biofilm. However, we found that the phrase kept its meaning even without this term and chose to remove it to avoid any ambiguity.
  3. “kinase … which sensed external stimuli”. Provide more mechanistic description. - we’ve added a more thorough description of the sensing-regulation process of this two-component system.
  4. “A good way to eradicate biofilms could be inhibition of pili formation”. How good is it? Are there comparative data? Specify the inferior approaches or revise” – we’ve found data suggesting this mechanism is an effective way of biofilm eradication, but not specifically on the Enterococcus We will continue to support this argument but we rephrased the whole paragraph, added some references to back up what was already researched up until now and inserted words and phrases suggesting probability and correlation for the notions we weren’t able to back up with the current available data.
  5. “Species-specific genes and virulence factors”. Are the gelatinase and cytolysin genes species-specific? Probably, particular gene variants are species-specific. – Yes, the cytolysin and gelatinase genes variants presented in the article are species-specific. We did not get in depth about the particular gene variants in the abstract because that is not the point of the abstract and we made the change acordingly.
  6. What is a difference between regulating gene expression and target gene activation? – We have rephased the sentence – we meant that quorum-sensing plays a crucial role in the gene expression but also in the modulation of its expression (gene regulation).
  7. Lines 28, 32, 33. Italics in the genus name. – We’ve made the recommended modifications.
  8. Line 48. is mainly comprised of --> comprises – We’ve made the recommended modifications.
  9. Line 56. Biofilm formation is mediated by quorum sensing. It is probably regulated, not mediated. – We’ve made the recommended modifications.
  • Lines 56-57. Check the multiple use of “regulated” and “mediated”. – We’ve rephrased the sentence for a clearer way of scientifically expressing our ideas.
  • Line 76. Write amino acids in full. Please explain the appearance of sprE – We’ve rephased – we meant to say that the gelatinase-coding gelE gene is part of an operon alongside the sprE gene and that this operon’s expression is regulated by the fsr quorum-sensing system.
  • Lines 62-63. Please cite those articles after the expression “the Enterococcus genus” – We’ve made the recommended modifications.
  • Line 81. In the introduction, the roles of gelatinase were not mentioned. – We’ve rephrased the sentence.
  • Line 97. Change pathogeny to pathogenicity. - We’ve made the recommended modifications.
  • Lines 145, 146. The old section title was not deleted. - We’ve made the recommended modifications.
  1. Figure 1 carries no information related to the main text and should be deleted with the respective note on the Line 145. It is now deleted
  • Figure 2. Mark phases on the image, and annotate the shown structures, including the elongated O-like structure in the middle of the cells. We’ve made the recommended modifications (Figure 2 has become Figure 1).

Comments on the Quality of English Language:

There is a need to correct English.

R: We have made various corrections throughout the paper where the grammar and spelling were indeed faulty.

Reviewer 2 Report

The article written by Schiopu et al. focuses on the factors involved in the biofilm formation ability of some species of Enterococcus. Although the topic is well known, the present review provides a beautiful summary of the literature. The writing style is captivating and the figures are, with some exceptions, explanative.

Specific comments:

1.     The abstract mention E. faecalis, and, as E. faecalis is the most common species in the genus, I can understand why is being mentioned here. However, it would be interesting to add a sentence about E. faecium too, as E. faecium poses more therapeutical challenges. Also, E. faecium is often mentioned in the text, hence please mention it in the abstract too.

2.     The aim of the study promises “the most complete description” (Line 63) of the biofilm formation steps. Please rephrase.

3.     Please increase the quality of Figure 3

4.     What is the link between C. albicans and Enterococcus spp.? Quorum sensing is a well-characterized process, in fungi, as well as bacteria. The authors included an extensive section about C. albicans quorum sensing in a review about Enterococcus spp. Please remove it or explain how are Lines 307-327 in accordance with the aim of the study?

5.     Lines 361-362 mention that “paracrine pheromone signaling pathway  shows similarities between Candida albicans and E. faecalis”. If the review also wants to focus on the interspecies relationships, please add other species too (as there are many studies available on the topic) or mention why did you chose to mention just this one (inclusion, exclusion criteria)

6.     Lines 467-493 explain the eDNA. While the subject is very interesting, it does not fit the aim of the study. The paragraphs can be substantially shortened.

7.     Numerous virulence factors are described in the manuscript, some for E. faecium, others for E. faecalis. A table, summarizing the information, would be helpful.

8.     Please also include information (at least shortly) about other Enterococcus spp. (as the title of the article refers to the Enterococcus genus.

9.     The abstract mentions the importance of understanding the biofilm formation process for further “strategies to combat multidrug-resistant infections”, but the idea is little discussed in the text. Line 617 mentions (again) that the paper “highlights potential methods for eradicating biofilms”. Besides “the inhibition of Pilli formation” (Line 190) by “implementing some types of vaccines”, the idea is not “highlighted in the text”. Please further develop the idea.

10.  The conclusion section is too big and it contains repeating information. Please rewrite it.

Line 28 - please italicise "Enterococcus;

"Species" can be abbreviated "spp."

Author Response

Dear reviewer no. 2:

Thank you for your extensive and thorough review. We have made the necessary adjustments. Below are our responses to each of the suggested modifications:

  1. The abstract mention E. faecalis, and, as E. faecalis is the most common species in the genus, I can understand why is being mentioned here. However, it would be interesting to add a sentence about E. faecium too, as E. faecium poses more therapeutical challenges. Also, E. faecium is often mentioned in the text, hence please mention it in the abstract too.

R: We mentioned E. faecium both in the abstract and the Introduction, highlighting it's increasing resistance to vancomycin.

  1. The aim of the study promises “the most complete description” (Line 63) of the biofilm formation steps. Please rephrase.

R: We rephrased the aim of the study to better reflect what was feasibly achieved in this narrative review.

  1. Please increase the quality of Figure 3

R: We revamped Figure 3, modified the shapes used, correcting one arrow, removed some misleading vectors, and used a more readable typeface (now Figure 2 since we deleted Figure 1).

  1. What is the link between C. albicans and Enterococcus spp.? Quorum sensing is a well-characterized process, in fungi, as well as bacteria. The authors included an extensive section about C. albicans quorum sensing in a review about Enterococcus spp. Please remove it or explain how are Lines 307-327 in accordance with the aim of the study?

R: We agree that we had made a distant analogy, and we have now shortened it substantially.

  1. Lines 361-362 mention that “paracrine pheromone signaling pathway shows similarities between Candida albicans and E. faecalis”. If the review also wants to focus on the interspecies relationships, please add other species too (as there are many studies available on the topic) or mention why did you chose to mention just this one (inclusion, exclusion criteria)

R: The aim of the review does not include interspecies relationships. We have rephrased said lines in a way that no longer includes the Candida analogy.

  1. Lines 467-493 explain the eDNA. While the subject is very interesting, it does not fit the aim of the study. The paragraphs can be substantially shortened.

R: The paragraphs explaining the various general functions of eDNA have now been shortened.

  1. Numerous virulence factors are described in the manuscript, some for E. faecium, others for E. faecalis. A table, summarizing the information, would be helpful.

R: We have now created a table in the Introduction section, summarizing the various factors and the identified functions, with references.

  1. Please also include information (at least shortly) about other Enterococcus spp. (as the title of the article refers to the Enterococcus genus.

R: We included information about other Enterococcus spp. at line nr 46-56

  1. The abstract mentions the importance of understanding the biofilm formation process for further “strategies to combat multidrug-resistant infections”, but the idea is little discussed in the text. Line 617 mentions (again) that the paper “highlights potential methods for eradicating biofilms”. Besides “the inhibition of Pilli formation” (Line 190) by “implementing some types of vaccines”, the idea is not “highlighted in the text”. Please further develop the idea.

R: We partially rephrased the abstract and the aim to downplay the therapeutic aspect, and to make it clear we only aimed to provide a few directions for future therapeutic research.

  1. The conclusion section is too big and it contains repeating information. Please rewrite it.

R: We have shortened the conclusion substantially.

Comments on the Quality of English Language

Line 28 - please italicise "Enterococcus;

R: it is now italicisized

"Species" can be abbreviated "spp."

R: we abbreviated "Enterococcus species" to "Enterococcus spp."